# The Influence of Food Environments on Food Security Resilience during the COVID-19 Pandemic: An Examination of Urban and Rural Difference in Kenya

**DOI:** 10.3390/nu14142939

**Published:** 2022-07-18

**Authors:** Emily V. Merchant, Tasneem Fatima, Alisa Fatima, Norah Maiyo, Vincent Mutuku, Susan Keino, James E. Simon, Daniel J. Hoffman, Shauna M. Downs

**Affiliations:** 1New Use Agriculture and Natural Plant Products Program, Department of Plant Biology, Rutgers University, New Brunswick, NJ 08901, USA; evr.merchant@rutgers.edu (E.V.M.); jimsimon@rutgers.edu (J.E.S.); 2Center for Agricultural Food Ecosystems, The New Jersey Institute for Food, Nutrition, and Health, Rutgers University, New Brunswick, NJ 08901, USA; dhoffman@sebs.rutgers.edu; 3Urban-Global Public Health, Rutgers School of Public Health, Newark, NJ 07102, USA; ktasneem243@gmail.com (T.F.); af703@sph.rutgers.edu (A.F.); 4Academic Model Providing Access to Healthcare (AMPATH), Eldoret 30100, Kenya; noramaiyo@gmail.com; 5G-thamini Youth Group, Nairobi, Kenya; vmmswahili@gmail.com; 6Department of Health Management & Human Nutrition, School of Public Health, Moi University, P.O. Box 4606, Eldoret 30100, Kenya; susankeino@gmail.com; 7Department of Nutritional Sciences, New Jersey Institute for Food, Nutrition, and Health, Center for Childhood Nutrition Education and Research, Rutgers University, New Brunswick, NJ 08901, USA

**Keywords:** agriculture, diets, food access, food insecurity, food availability, subsistence farming

## Abstract

Hunger and food insecurity has worsened due to the COVID-19 pandemic. The types of food environments (e.g., natural/built) that people can access may improve household resilience to food-system shocks. This paper examines (1) urban and rural differences in the perceived influence of the COVID-19 pandemic on agricultural, livelihoods, food environment attributes, diets; and (2) whether access to different food environments was associated with food security. A two-part telephonic survey (COVID-19 Surveillance Community Action Network Food Systems Tool and Household Food Insecurity Access Scale) was conducted in Western Kenya (*n* = 173) and an informal settlement in Nairobi (*n* = 144) in January/February 2021. Limitations on the acquisition of farm inputs and movement restrictions had an adverse impact on agriculture and food sales. Urban residents reported a more significant impact on livelihoods (97% vs. 87%, *p* < 0.001), with day laborers being the most impacted. Rural respondents reported access to significantly more food environments and lower food insecurity. Multiple linear regression analysis revealed that younger respondents, ≤1 income source, had more difficulty acquiring food, decreased access to cultivated environments, and increased access to informal markets were predictors for higher food insecurity. These data indicate that access to specific types of food environments may improve household resilience.

## 1. Introduction

The prevalence of hunger has been steadily increasing, particularly in Africa, Latin America, and Western Asia, since 2017 [1]. In addition, approximately 25% of the world’s population is classified as food insecure, where unreliable access and availability of nutritious foods can increase the risk of malnutrition [2]. Recent reports estimate that between 2019 and 2020, hunger increased by 161 million and food insecurity increased by 320 million individuals, due in large part to the Coronavirus Disease 19 (COVID-19) pandemic, with the largest increases in undernutrition seen in Africa and Asia [2].

It is well documented that the COVID-19 pandemic has influenced diet quality and quantity, particularly among vulnerable populations in low- and middle-income countries through several causal pathways within the food environment [3,4]. Food environments include the natural (e.g., wild and cultivated) and built (e.g., informal and formal markets) spaces that consumers interface with to access food. Disruptions to the global food supply may limit the availability and quality of nutritious, culturally preferred foods within the built food environment (e.g., kiosks, mobile vendors, and supermarkets), particularly in the informal market space [5]. Furthermore, mitigation strategies, such as government-mandated lockdowns, a policy used to reduce the spread of the virus by requiring individuals to stay in their homes for a determined period (often 2 weeks), impeded economic livelihood through decreased employment and income-generating activities as well as an individual’s ability to physically access food in the built food environment [6]. Alternatively, access to cultivated or wild food environments for farming or foraging may increase the accessibility and availability of nutritious foods, particularly during food shocks. However, different regions have unique conditions that may worsen the impacts of the COVID-19 pandemic, and further influence the food environments, such as the impact of the desert locust (*Schistocerca gregaria*) outbreak on agriculture in Kenya that devastated crops and reduced the availability of food.

Country-specific mitigation strategies could further contribute to food security. The first case of COVID-19 in Kenya was reported in mid-March of 2020. Consistent with World Health Organization (WHO) guidelines, the Kenyan government immediately instituted several restrictions to mitigate disease spread such as school closures, a work-from-home order, and travel restrictions [7]. In addition, to soften the financial stress of the pandemic, the Kenyan government enacted several programs such as cash transfers and job creation programs focused on urban communities. Despite these efforts, results of research revealed that the COVID-19 pandemic disproportionally affected urban residents in Kenya, who reported a more significant influence on income and diets [8,9] than those in other peri-urban and rural areas. Furthermore, prior to the COVID-19 pandemic, food insecurity was already increasing in Kenya. *The State of Food Security and Nutrition in the World 2021* [2] reported an increase in severe food insecurity from 17% (8.3 million individuals) to 25% (13.5 million individuals) between the 2014–2016 and 2018–2020 census. Food system shocks, such as the COVID-19 pandemic are expected to further impact food insecurity within these vulnerable populations.

While various studies have examined the impact of the COVID-19 pandemic on food security at the household and national level, there is limited information on how access to different types of food environments may alter the influence of the COVID-19 pandemic on food security [4,8]. Exploring how the influence of the COVID-19 pandemic differed in urban and rural communities, and subsequently, access to different food environments in Kenya, can provide valuable insight into the drivers of food insecurity in these communities. The purpose of this study is to inform policy on appropriate response measures to withstand future and potentially perpetual food system shocks [10,11]. Rapid assessments, such as the COVID-19 Surveillance Community Action Network for Food Systems (C-SCAN) [12] and the Household Food Insecurity Access Scale (HFIAS) [13], can be used to identify resilience and vulnerabilities within the food system and food security status, respectively, particularly in low- and middle-income countries, which are disproportionately impacted by food system shocks.

This study used a two-part telephonic study comprising of the C-SCAN and HFIAS survey tools and was conducted in urban and rural Kenya. This study aimed to examine urban and rural differences in the perceived influence of COVID-19 on (1) agricultural production and livelihood; and (2) food environment attributes (food access, accessibility, price) and diets. Furthermore, this paper examined whether access to different environments (e.g., wild, cultivated, informal, and formal) impacted food security. We hypothesize that during the COVID-19 pandemic, urban residents will report decreased access to both the natural and built food environment negatively influencing food security. 

## 2. Materials and Methods

### 2.1. Overview of Study and Study Setting

This was a household (HH) survey conducted telephonically in January and February 2021 in two regions of Kenya: smallholder farmers in rural communities in Western Kenya and the urban “informal settlement” Mukuru in the country’s capital, Nairobi. In rural Kenya, agriculture contributes to 60% of employment, which can provide access to available and affordable crops and livestock [14]. While a majority of Kenya is still rural, it is rapidly urbanizing, subsequently increasing the emergence of informal settlements [15]. Informal settlements in Kenya, such as Mukuru, are characterized by inadequate access to safe water and sanitation, overcrowding, and insecure land tenure [16].

### 2.2. Sample Population

Two distinct areas of Kenya were approached for this study: rural communities in Western Kenya and an urban informal settlement. A convenience sample, of rural and urban households, was used due to the need to conduct a rapid assessment during the pandemic. For the HH survey, the study population included adults (18 years of age and older) who either resided in Mukuru or within five counties of Western Kenya: Bungoma, Busia, Kisumu, Trans Nzoia, and Uasin Gishu. These areas were selected based on prior work conducted by members of the study team [17,18,19,20,21]. The total sample size (*n* = 300) was selected based on sample size calculations conducted by similar researchers using the same rapid assessment (C-SCAN) in other geographical locations [12,22]. In order to reach the total sample size in both settings, a larger study sample was approached for participation in the study. 

In the rural communities, HHs across the five counties (*n* = 300) were randomly selected using a random number generator from a list of households (*n* = 500) that participated in a USAID project led by the study team [18,19]. A member of the study team (N.M.) called each of the randomly selected HHs to introduce them to the present study and request permission to call them in the future for participation in the study. Rural recruitment ceased once 150 individuals were surveyed. In the urban setting, a member of the study team (V.M.), who works in Mukuru, recruited individuals, and ascertained their phone numbers. Urban recruitment ceased once 150 individuals were surveyed. Study participants were not compensated.

### 2.3. Survey Instrument

Questionnaires in all forms (in-person, electronically, telephonically, and by email) have been used to ascertain data related to aspects of food security such as perceptions and behavior relative to food waste [23]. This study utilized a two-part survey that contained a tool titled COVID-19 Surveillance Community Action Network (C-SCAN) for Food Systems [12] and the Household Food Insecurity Access Scale (HFIAS) [13]. The C-SCAN survey is a transferable tool designed by Ahmed and colleagues [12] to evaluate the influence of food environments on food security resilience during the COVID-19 pandemic and has been used in India [22] and the Northern Great Plains in the United States [24]. The survey contained six parts and elicited information regarding socio-demographic profiles, food sources, perceptions of food security aspects such as food availability, access, diets, household income, and farming and gardening systems. From a list of commonly consumed foods in Kenya, respondents were asked which foods were easier or harder to access; which foods increased or decreased in price; and which foods they consumed more or less of. These responses were then totaled based on food groups. A majority of the questions in the survey tool were binary (Yes/No) responses, which allowed for rapid assessment and analysis. In addition, select questions were open-ended to allow for additional context (e.g., explain why you are consuming more or less of a specific food). The HFIAS questionnaire included nine questions to ascertain the level of food security within households [13]. An overall HFIAS score was calculated (0–27), which represents the degree of household food insecurity in the past 4 weeks. A higher HFIAS score denotes a higher degree of food insecurity. In addition, households were categorized based on food security levels: (1) food-secure, (2) mildly food-insecure, (3) moderately food-insecure, and (4) severely food-insecure based on calculations provided in the toolkit. 

### 2.4. Data Collection 

The survey tool was incorporated into an online survey platform for ease of collection and analysis. The survey instrument was translated from English into Swahili by a member of the research team (N.M.). In both study settings, a study member (N.M.) called each selected HH to schedule a date and time to conduct the telephonic survey. Data was collected by four individuals, one member of the study team (N.M.) and three trained enumerators. The enumerators conducted the survey in a standard setting (either an office at Moi University or AMPATH); however, it was not required that interviewees be in a standard setting. To keep methods consistent between enumerators, the interview was not adjusted for the interviewee outside the use of probing questions as dictated by the survey tool. To correct for potentially falsified data, the data were routinely audited, and outliers were removed from analysis. 

### 2.5. Data Analysis 

Descriptive statistics (means and frequencies) were used to summarize household demographics, food environment attributes, and HFIAS score. Independent *t*-tests were used to examine continuous variables and χ^2^ test was used for comparison between categorical variables. An ANOVA test was used to examine the association of HFIAS scores with age groups. A Tukey post hoc ANOVA test was used to determine the statistical differences between groups. Open-ended questions were aggregated, and responses were selected based on relevancy and thoroughness of the response to provide context to the quantitative data. 

Multiple linear regression analyses were used to determine factors that contributed to the HFIAS score. HFIAS score was entered as the dependent variable. Socio-demographic factors (e.g., county type, age, and gender) and food environment factors (e.g., household changed diet and increased access to the cultivated environment) were entered as independent variables. Forward stepwise analysis was conducted to determine statistically significant independent variables. The final model included statistically significant independent variables and variables that were predicted to have an effect but were not determined statistically significant in the forward stepwise analysis. This ensured that aspects relative to each of the research aims were analyzed in the regression model. Nominal and ordinal variables were recoded into dichotomous dummy variables. The final model included one linear independent variable (age) and nine independent dichotomous dummy variables (1. County type 1 = urban, 0 = rural; 2. Gender 1 = women, 0 = men; 3. More than one income 1 = yes; 4. Practice farming 1 = yes; 5. Acquiring food more difficult compared to pre-COVID 1 = yes; 6. Increased access to cultivated food environment 1 = yes; 7. Increased access to informal food environment 1 = yes; 8. Reported change in food price 1 = yes; 9. Household change diet 1 = yes). Multiple linear regression analyses were conducted for the overall study sample and individually for each county setting. All statistical analyses were conducted using SPSS (IBM SPSS Statistics v 26; Armonk, NY, USA) and a *p* value of <0.05 was used to denote statistical significance. 

### 2.6. Ethical Considerations

Ethical approval in the United States was provided by the Institution Review Board at Rutgers University, the State University of New Jersey. Ethical approval in Kenya was provided by the Moi University and Moi Teaching and Referral Hospital’s Institutional Research and Ethics Committee. All study participants provided informed oral consent to participate in the study and for use of the data in publications. The use of oral consent was approved by the ethical review boards due to minimal associated risk, low literacy rates among the study population, and limitations from a telephonic study method.

## 3. Results

### 3.1. Demographic

A total of 353 individuals participated in the C-SCAN; however, due to incomplete datasets, 36 samples were omitted from the analysis. A total of 317 samples were included for analysis (*n* = 144 urban and *n* = 173 rural). Table 1 reports the demographic characteristics of the sample population. Overall, respondents were female (85%) and were, on average, 40 years old. The rural population was significantly older compared to the urban population (*p* < 0.001).

Overall, a majority of individuals (71%) obtained only one source of income with the sale of food items (82%) and day laborer (51%) representing the highest reported forms of employment in rural and urban communities, respectively. Most farmers lived in rural areas (97%; *p* < 0.001). Overall, mixed farming was the most commonly reported form of agriculture (78%), defined as growing crops and rearing livestock on a small to medium size piece of land for sale or home consumption. Other reported farming types included subsistence (growing crops for only household consumption) and arable farming (growing of only crops) (32% and 8%, respectively). In addition, rural households had access to significantly more food sources than urban households (1.9 ± 0.7 vs. 1.4 ± 0.6; *p* < 0.001).

### 3.2. Influence of COVID-19 on Agricultural Practices and Livelihood

#### 3.2.1. Agricultural Practices

Approximately a quarter of the farmers reported changing the crops that they grew due to the coronavirus outbreak. While some farmers noted shifting crops from one type to another, a few farmers noted more general changes such as shifting cultivation methods as summarized by a rural participant, “changed to crops that take shorter time to be ready like vegetables and tomatoes.” In addition, to meet market demand, some farmers “grew varied types of vegetables on a larger piece of land” (rural). Moreover, many farmers reported either starting to or increasing chicken rearing. In addition, some farmers noted that their farming practices changed due to labor shortages as summarized by a rural respondent, “started using chemicals to control the weeds in the farm due to lack of laborers….” Additional context on how laborer shortages, and their relative cost, impacted farming practices was provided by a rural respondent: “reduced the farming land as there was lack of labor which was quite expensive.” In addition to the COVID-19 pandemic, many farmers noted that drought, heavy rains, and plant diseases negatively impacted agricultural production. Figure 1 summarizes the specific ways that the pandemic influenced agricultural practices as well as concerns over future impact.

Many farmers reported how decreased transportation and movement restrictions impacted their agricultural practices, which decreased access to farm inputs (60%) and decreased their ability to sell crops (63%). The acquisition of farm inputs was also influenced by the farmers’ low purchasing power, which subsequently influenced production and sale of crops as summarized: “fertilizer is expensive to acquire and planted crops with lower amounts than recommended rates. The production output was low and not enough to supply to the market” (rural). Relative to the sale of crops, many farmers noted that during the lockdown to “avoid [getting] infected with COVID-19 you [sold] the produce in the farm” (rural). Some respondents also noted “selling from home because of restrictions to go to the marketplaces by the government” (rural). In some instances, this was noted as a positive attribute as the “buyers could easily come to the homestead as opposed to the market due to movement restrictions” (rural). However, not all those who sold food crops were farmers, and, therefore, did not have a farm to serve as a food outlet.

Participants reported that the lockdown impacted their ability to sell produce as “marketplaces were closed down” (rural). Furthermore, when travel restrictions were lifted, some participants noted that “the market was flooded with food stuffs and buyers were less…” (rural). In addition, clients’ “…low purchasing power and low selling prices” (rural) impacted sales. Moreover, one respondent noted that ready customers weren’t always available as “…most buyers had moved from the town to rural areas” (rural). Often in response to market demand, over a quarter (27%) of respondents reported that they changed the types of foods they were selling: “harvest and sold maize and beans as their market prices were high to obtain income to buy other family needs” (rural). In addition, some respondents noted that they decreased what they sold at the market because the “whole family was home and the high consumption hence little left for sale” (rural). Moreover, a majority of farmers noted that their ability to sell crops (58%) was a major concern for how the COVID-19 pandemic would continue to impact their livelihoods.

#### 3.2.2. Livelihood

Overall, 92% of the participants reported a change in household income due to the COVID-19 pandemic. However, significantly more urban respondents reported a change in livelihood compared to rural participants (97% urban and 87% rural *p* = 0.005). In the rural communities, food vendors reported the highest changes in income (82%) compared to day laborers (51%) in the urban community (Figure 2). Due to changes in household income, some respondents noted having to take on additional work as summarized by an urban respondent: “from sack gardening to selling clothes in the evening.”

### 3.3. Perceived Influence of the COVID-19 Pandemic on Food Environment Attributes

#### 3.3.1. Access to Different Types of Food Environments

Prior to the pandemic, the most reported food environment for rural communities was cultivated places compared to informal markets in rural communities. A minority of respondents indicated that the locations where they acquired their food had changed since the COVID-19 pandemic (35%), with significantly more urban respondents reporting a change compared to rural respondents (44% and 27%, respectively, *p* < 0.001). Table 2 summarizes how access to different types of food environments shifted due to the COVID-19 pandemic. Urban respondents reported decreased access to formal markets, supplemental food, and cultivated spaces, as well as increased access to informal markets. Rural communities reported decreased access to formal markets and increased access to cultivated places. In addition, roughly 13% of rural respondents reported decreased access to informal markets while 13% reported increased access. Moreover, rural communities reported a significantly higher variety of different food environments: 69% of the rural community reported two food environment types compared to 68% of urban respondents reported access to only one type of food environment (*p* < 0.001) (Figure 3).

#### 3.3.2. Food Accessibility and Price

Overall, 85% of participants reported that it was harder to obtain food since the beginning of the pandemic; however, there was a significant difference between the reporting in urban and rural communities (Figure 4). A quarter of the rural respondents reported that their ability to get food was about the same as before the COVID-19 pandemic compared to 1% of the urban community (*p* < 0.001). This was summarized by one urban participant who reported that “all types of foods” were harder to acquire, meanwhile a rural participant reported that “[food] was not harder to get but the money was scarce.”

Table 3 depicts the accessibility, price, and consumption of different food groups among urban and rural households. Ninety-one percent of participants indicated that food prices changed due to COVID-19. In both the rural and urban communities, respondents indicated difficulty accessing, as well as increased price of, animal-source proteins and grains, white roots, and plantains. The urban community also expressed difficulty accessing, and increased price of, dark leafy vegetables, other fruits and vegetables, and vitamin A-rich fruit and vegetables. While a small portion of rural respondents noted that some food groups were easier to acquire, this was not reflected in the urban community. Moreover, over a third of the participants reported that no foods decreased in price. In addition, in some cases, the products sold were further altered as summarized an urban respondent, “none of the foods decreased in price but the quantity was reduced, and the prices remained the same or it increased.”

#### 3.3.3. Diets

Figure 4 summarizes the influence of the COVID-19 pandemic on various components of diets and diet-related concerns in urban and rural Kenya. In both settings, food affordability was reported as the major concern (61% rural and 99% urban; *p* < 0.001). In addition, significantly more urban respondents reported that it was harder to obtain cooking fuel relative to the rural communities (96% and 36% respectively, *p* < 0.001). Several rural households reported that acquiring cooking fuel remained the same due to their ability to “obtain from trees around the farm.” However, some rural respondents did report a change in access to cooking fuel as summarized: “prices of buying charcoal increased and it was scarce.”

Significantly more urban respondents noted a change in diet compared to the rural respondents (60% and 37% respectively, *p* < 0.001) (Table 4). Table 3 reports the specific foods that their households consumed more or less of since the start of the COVID-19 pandemic. Rural communities reported increased consumption of dark leafy greens and other fruits and vegetables while there was a reported decrease in grains, white roots, and plantains, as well as and animal-source protein. In the urban communities, there was a reported increase in consumption of other fruits and vegetables while there was a reported decrease in animal-source proteins; grains, white roots, and plantains, as well as vitamin A-rich fruits and vegetables. In addition, a majority of participants (58%) reported that they began to consume medicinal plants (Table 4). The most commonly consumed medicinal plants were African Indigenous Vegetables, most commonly African nightshade, and a medicinal drink called Dawa, which were most often acquired at the local markets or kiosks (Figure 5).

### 3.4. Food Security

The analysis revealed that setting, socio-demographic characteristics, and access to a variety of different types of food environments contributed to HFIAS scores. Table 5 reports HFIAS score by setting and socio-demographic characteristics. The mean HFIAS score for urban households was double that of rural households (18.1 ± 3.3 vs. 9.0 ± 5.4, *p* < 0.001). In the urban setting, the higher HFIAS score is attributed to both higher reported events that contribute to food security, as measured by responses to the HFIAS scale (e.g., yes/no), as well as higher frequency of these events (Figure 1). Furthermore, female respondents were reported to be significantly more food insecure compared to the male population (14.2 ± 6.2 vs. 8.0 ± 5.7; *p* < 0.001). There was also a significant association between food security and age (*p* < 0.001), where the highest scores were found among the youngest population (18 to 24 years old, 17.9 ± 4.1), while the lowest scores were found among the older populations (55 to 64 years old, 8.4 ± 6.8 and 65 years and older, 10.0 ± 6.6).

Results of the multiple linear regression analysis are summarized in Table 6. Independent of setting specific analysis, residing in an urban community, being younger, having one or less sources of income, difficulty acquiring food, decreased access to cultivated food environments, and increased access to the informal markets were predictors for a higher HFIAS score. When the analysis was performed for each setting, the model revealed that in both settings, difficulty acquiring food was a predictor for a higher HFIAS score; however, being younger and having decreased access to the cultivated food environments were also predictors of a higher HFIAS score in the rural setting but had no significant influence in the urban setting. None of the models revealed significant results for gender, practicing farming, reported change in food price, or change in diet in any of the settings.

## 4. Discussion

The rapid assessment tool C-SCAN and the HFIAS survey tool revealed that the COVID-19 pandemic negatively influenced agricultural production and livelihoods, as well as food accessibility, availability, price, and consumption in urban and rural Kenya. In addition, the study revealed that depending on setting (e.g., urban or rural), access to specific food environments contributed to HFIAS scores. These findings provide insight into how access to specific types of food environments may improve household resilience to food system shocks.

### 4.1. Agricultural Production and Livelihoods

#### 4.1.1. Agricultural Production

Farmers were largely from the rural communities and practiced mix farming (growing crops and rearing livestock). Respondents reported several ways in which the COVID-19 pandemic influenced agricultural production and sales through limited accessibility (e.g., individual purchasing power and travel restrictions), affordability (e.g., high labor wages), or availability of farm inputs and labor, which has also been reported on a global scale [25]. Rahimi and colleagues [26], reported that the pandemic’s influence on agriculture was partially a result of both domestic and global value-chain disruptions and movement restrictions, which decreased employment and farm revenue impacting purchasing power. In response to the pandemic, farmers and food retailers reported several ways that they altered their agricultural practices. While some practices had a neutral or even positive effect (e.g., selling produce from the farm), when the markets re-opened, the limited purchasing power and flood of market goods was reported to decrease overall sales. This further impacted farmers’ livelihood, which was expressed as a major concern for future sales.

Kenya has two distinct rainy seasons, March to May and October to December. The data for this paper were collected in January and February. The proximity between the rainy season and the data collection could have influenced responses. In addition, this may contribute to the reported environmental stressors that further influenced farm production, often resulting in reported crop failure. It is of note however, that climate change is reported to influence crop production, regardless of season; therefore, some of the stressors reported by farmers may be due to the increasing stress of climate change rather than seasonality, or a combination of the two [27].

#### 4.1.2. Livelihoods

Significantly more urban respondents reported a change in livelihood since the start of the COVID-19 pandemic. The loss of daily wage jobs was attributed to movement restrictions, which prohibited an individual’s ability to work outside of the home. Osiki [28] reported that the COVID-19 mitigation efforts of the Kenyan government caused a greater disruption of income sources in urban communities despite government stimulus and job creations [7,8,9]. The urban community reported a greater influence on livelihoods, which may be attributed to the severity of the lockdown policies in Nairobi, Kenya relative to the rural communities. Urban centers in Kenya were reported to have more stringent lockdown policies compared to its rural counterpart and relative to other African countries. In their study, Maredia and colleagues [9] found that impact on livelihoods was not significantly different between urban and rural settings, except for in Kenya and Nigeria. This aligns with the stringency and length of the COVID-19 mitigation policies in the different settings in each country.

In the rural communities, the COVID-19 movement restrictions influenced the availability of markets for retail sale, the farmers’ ability to transport their agricultural produce to the market, and the customers’ ability to access the markets. This, in turn, impacted the livelihoods of food vendors, and further disrupted the local value chain. Furthermore, limitations in refrigeration and restricted transportation resulted in large quantities of food waste [29], in turn limiting the availability of nutritious food to informal and formal markets [30]. The influence of the COVID-19 pandemic on livelihoods further exacerbated the prevalence of food insecurity within these vulnerable populations.

### 4.2. Food Environment Attributes and Diet

Despite the reported changes and restrictions to movement, rural households acquired food from a larger variety of food environments when compared to urban households. Furthermore, rural respondents reported fewer changes to the different types of food environments they had access to compared to the urban respondents. This may be due to the stringency of the lockdown period in urban Kenya compared to rural as well as the community’s primary food environment prior to the pandemic. A majority of rural households were farmers, and, therefore, already had access to the cultivated and wild food environment, which was less affected by the shelter-in place and travel restriction policies. On the contrary, the urban communities primarily received their food from the informal and formal market, which was severely impacted by the COVID-19 mitigation policies. The COVID-19 shelter-in-place policies limited food environment access, food accessibility, price, and diet quality [26,29].

Respondents in both settings reported changes in the accessibility and price for specific food commodities. Following a supply and demand relationship, food commodities that were reported to be harder to acquire, such as grains, white roots, and plantains, also had a reported price inflation, and lower consumption. Similar food price and consumption patterns have been observed pre-pandemic, particularly amongst low- and middle-income countries [31,32,33]. In addition, these studies report a shift in dietary consumption towards low-cost, highly processed foods. This has important implications towards policy level change during times of price shocks and disturbances to the global food supply chain, which impact the accessibility and price of nutritious foods.

Food environment attributes influenced diets in both the urban and rural settings. In the rural communities, higher reported food accessibility and subsequent consumption of indigenous foods, vegetables, and fruits, may be attributed to access to the natural food environment and relative stability in food expenditures. The natural food environment, which provides land for subsistence farming and foraging, has been reported to contribute to improved dietary diversity [34,35]. Furthermore, a study by Janssens and colleagues [36] in rural Kenya, found that food spending remained at pre-COVID levels, while school and transport spending decreased, which could contribute to food consumption patterns remaining relatively stable through the COVID-19 pandemic.

The accessibility and price of specific commodities were influenced by their relative supply chains. Food groups, such as animal-source protein, rely on domestic or global value chains, as well as external factors, such as refrigeration. Disruptions within these value chains resulted in food waste and dumping further influencing supply and price [26]. On the contrary, some food groups, such as dark leafy greens depended on a shorter, more local, supply chain and were reported to be easier to acquire and had a higher reported consumption particularly in the rural communities. Similar studies, which used a variation of the C-SCAN survey, found that households in China and India that relied primarily on cultivated food environments, and, therefore, shorter more local supply chains, were more resilient to negative changes in the food system and had to change their diet less to accommodate for those negative changes [12,22]. In addition to an impact diet quality, the COVID-19 pandemic had a large impact on food security.

### 4.3. Food Security

HFIAS scores in urban areas were almost double those reported in rural areas, suggesting a disproportionate influence of the pandemic on household hunger in the urban communities. Other studies found that informal urban settlements had higher food insecurity when compared to rural communities [4,37]. For example, a study by Pinchoff and colleagues [8] found that three-quarters of all study participants living in informal settlements in Nairobi, Kenya, skipped a meal in the previous week due to the COVID-19 pandemic. Furthermore, women were disproportionally impacted, with a significantly higher HFIAS score relative to men; yet sex was not a predictor of higher HFIAS score in the multiple linear regression analysis. Lastly, younger individuals had a significantly higher HFIAS score relative to older age groups. While prior studies indicated that age of household head was negatively associated with household food security pre-pandemic [38,39], during food system shocks, younger age groups may have developed fewer coping strategies as compared to the older age groups, which may have mitigated the influence of the COVID-19 pandemic on food security.

In addition to demographic characteristics, food environment attributes influenced HFIAS score. Difficulty acquiring food, decreased access to cultivated food environments, and increased access to the informal market were predictors for a higher HFIAS score. Participants who reported increased access to informal markets, yet higher HFIAS scores may be indicative of individuals who depended on the built food environment and therefore had to travel to an informal market, rather than rely on their own cultivated environment. Alternatively, and aligned with a prior study [12], access to the cultivated environment was predictive of a lower HFIAS score. There is a need for policies that increase access to the cultivated food environment in both rural and urban settings and invest in infrastructure for the local supply chains to mitigate the influence of food system shocks on food security.

### 4.4. Limitations

While this study has several strengths, such as study design and ascertaining new information relative to the influence the COVID-19 pandemic on food security in urban and rural communities in Kenya, there were also limitations. The data were solely self-reported and, therefore, accuracy is dependent on the participant’s recall ability. In addition, the study utilized a telephonic mode for data collection, which could have resulted in respondent fatigue and influenced responses. Furthermore, even though the study team in the field spoke the local languages, there may have been information lost due to the language and cultural barrier. Finally, most of the participants (85%) in the study were females, which may or may not have skewed the data.

## 5. Conclusions

This paper examined the perceived influence of the COVID-19 pandemic on food security and food environment attributes in urban and rural Kenya. The findings revealed that restrictions in movement negatively impacted agricultural production, sales, and food access across settings. In addition, disruption to domestic and global value-chains negatively impacted agricultural inputs and food availability and price. Farmers reported mitigation strategies to improve agricultural production; however, in some cases, decreased agricultural production was further exacerbated by environmental conditions such as weather. Despite these challenges, the rural community reported access to a higher diversity of food environments and a significantly lower Household Food Insecurity Access Scale (HFIAS) score compared to the urban communities. Moreover, access to the cultivated food environment was predictive of a lower HFIAS score. Future research directions include evaluating the influence of seasonality on food availability and access in the natural food environment during food system shocks and surveying additional geographical areas and informal settlements in Kenya. Future policies and interventions should focus on increasing household access to the natural food environment when available to mitigate the impact of food system shocks, particularly for informal settlements. This can be accomplished by partnering urban agricultural initiatives with abandoned lots and grounds of local establishments and businesses such as schools and churches. Furthermore, increasing food availability and access during food system shocks can mitigate food insecurity during these times.

## Data Availability

The data presented in this study are available on request from the corresponding author.

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
