# Peer review of "The Influence of Food Environments on Food Security Resilience during the COVID-19 Pandemic: An Examination of Urban and Rural Difference in Kenya"

_nutrients, 2022, doi:10.3390/nu14142939_

Round 1

Reviewer 1 Report

The manuscript represents an interesting practical analysis on influence of food environments on food security resilience during the COVID-19 Pandemic, based on examination of urban and rural difference in Kenya. The method described in the manuscript has been adequately chosen and presented. Interpretation of the results and conclusions drawn from the results are adequate. Authors’ statements are well supported with relevant references, including their recent surveys. 

Author Response

The manuscript represents an interesting practical analysis on influence of food environments on food security resilience during the COVID-19 Pandemic, based on examination of urban and rural difference in Kenya. The method described in the manuscript has been adequately chosen and presented. Interpretation of the results and conclusions drawn from the results are adequate. Authors’ statements are well supported with relevant references, including their recent surveys. 

Our Response: Thank you for the kind and positive comments and review.

Reviewer 2 Report

Comment: Thank you for the opportunity to read the manuscript entitled “The Influence of Food Environments on Food Security Resilience During the COVID-19 Pandemic: An Examination of Urban and Rural Difference in Kenya”. The manuscript investigates: (a) the urban and rural differences in the perceived influence of the COVID-19 pandemic on agricultural, livelihoods, food environment attributes, diets; (b) whether access to different food environments is associated with food security. I have greatly appreciated the efforts of the authors to investigate a topical concern in Kenya. The manuscript is well-written and well-structured, and the materials and methods have been in-depth described. Results are clear and comprehensive, as well as discussions. There are some minor revisions need to be addressed.

Introduction: “Introduction” is clear and comprehensive. 

Lines 65-74. I suggest the authors adding some more statistics and facts related to the COVID-19 pandemic in Kenya, as well as its consequences on food security in Kenya. Is there the chance to include some statistics related to the percentage (and absolute value) of undernourished or food insecure people in Kenya? And before and after the COVID-19 pandemic? It would be interesting and would increase scientific soundness and originality of the research. 

Lines 75-86. I suggest the authors describing the materials and methods (although briefly) adopted in the research. 

Materials and Methods: “Materials and Methods” are clear and comprehensive. The authors describe the sampling strategy, the data collection process and the data analysis. If the authors are willing to, they could add a brief statement related to the use of surveys in investigating food security during the COVID-19 pandemic. Here is an interesting article, which could be cited to strengthen the adoption of such a method to collect reliable and useful data. 

Amicarelli, V., Lagioia, G., Sampietro, S. and Bux, C. (2022). Has the COVID-19 pandemic changed food waste perception and behavior? Evidence from Italian consumers. Socio-Economic Planning Sciences, 82, Part A, 101095. 10.1016/j.seps.2021.101095

Further, in developing the survey, have the authors developed some research questions or research hypotheses? 

Results: “Results” are clear and comprehensive. The authors have done great efforts to provide synthetic and useful data. Either Tables or Figures are clear. The authors have provided both quantitative and qualitative results and have avoided redundancies. 

Discussion: “Discussion” are clear and comprehensive. The authors have developed critical outcomes based on previous results. 

I suggest the authors adding some more future research directions. 

Author Response

Comment: Thank you for the opportunity to read the manuscript entitled “The Influence of Food Environments on Food Security Resilience During the COVID-19 Pandemic: An Examination of Urban and Rural Difference in Kenya”. The manuscript investigates: (a) the urban and rural differences in the perceived influence of the COVID-19 pandemic on agricultural, livelihoods, food environment attributes, diets; (b) whether access to different food environments is associated with food security. I have greatly appreciated the efforts of the authors to investigate a topical concern in Kenya. The manuscript is well-written and well-structured, and the materials and methods have been in-depth described. Results are clear and comprehensive, as well as discussions. There are some minor revisions need to be addressed.

Our Response: Thank you for the kind review. The article was reviewed for general spelling and grammatical errors and corrections were made using track changes in the Word file and the requested minor revisions were addressed.

Introduction: “Introduction” is clear and comprehensive. 

Lines 65-74. I suggest the authors adding some more statistics and facts related to the COVID-19 pandemic in Kenya, as well as its consequences on food security in Kenya. Is there the chance to include some statistics related to the percentage (and absolute value) of undernourished or food insecure people in Kenya? And before and after the COVID-19 pandemic? It would be interesting and would increase scientific soundness and originality of the research. 

Our Response: The introduction now includes data relative to percentage and absolute value of food insecure individuals as reported in the FAO, IFAD, UNICEF, WFP, and WHO (2021). The State of Food Security and Nutrition in the World 2021. Transforming food systems for food security, improved nutrition and affordable healthy diets for all. FAO https://doi.org/doi.org/10.4060/cb4474en.

“Prior to the COVID-19 pandemic, food insecurity was already increasing in Kenya. The State of Food Security and Nutrition in the World 2021 reported an increase in severe food insecurity from 17% (8.3 million) to 25% (13.5 million) between the 2014-2016 and 2018-2020 census. Food system shocks, such as the COVID-19 pandemic are expected to further impact food insecurity within these vulnerable populations.” Lines 74-79

Lines 75-86. I suggest the authors describing the materials and methods (although briefly) adopted in the research. 

Our Response: A short description of the materials and methods were added to the introduction.

“Rapid assessments such as the COVID-19 Surveillance Community Action Network for Food Systems (C-SCAN) and the Household Food Insecurity Access Scale (HFIAS) can be used to identify resilience and vulnerabilities within the food system and food security status, respectively, particularly in low- and middle-income countries, which are disproportionately impacted by food system shocks.

This study used a two-part telephonic study comprising of the C-SCAN and HFIAS survey tools and was conducted in urban and rural Kenya.” Lines 87-94

Materials and Methods: “Materials and Methods” are clear and comprehensive. The authors describe the sampling strategy, the data collection process and the data analysis. If the authors are willing to, they could add a brief statement related to the use of surveys in investigating food security during the COVID-19 pandemic. Here is an interesting article, which could be cited to strengthen the adoption of such a method to collect reliable and useful data. 

Amicarelli, V., Lagioia, G., Sampietro, S. and Bux, C. (2022). Has the COVID-19 pandemic changed food waste perception and behavior? Evidence from Italian consumers. Socio-Economic Planning Sciences, 82, Part A, 101095. 10.1016/j.seps.2021.101095

Our Response: This comment is particularly helpful. Thank you for the suggestion and article. This article, as well as an expansion of discussion around the C-SCAN survey tool has been added to the manuscript as follows.

“Questionnaires in all forms (in-person, electronically, telephonically, by email) have been used to ascertain data related to aspects of food security such as perceptions and behavior relative to food waste [25]. This study utilized a two-part survey that contained a tool titled COVID-19 Surveillance Community Action Network (C-SCAN) for Food Systems [12] and the Household Food Insecurity Access Scale [13]. The C-SCAN survey is a transferable tool designed by Ahmed and colleagues [12] to evaluate the influence of food environments on food security resilience during the COVID-19 pandemic and has been used in India [23] and the Northern Great Plains in the United States [24].” Lines 143-151

Further, in developing the survey, have the authors developed some research questions or research hypotheses? 

Our Response: Yes, the study’s primary hypothesis was added to the introduction.

“We hypothesize that during the COVID-19 pandemic, urban residents will report decreased access to both the natural and built food environment negatively influencing food security.”  Lines 97-101

Results: “Results” are clear and comprehensive. The authors have done great efforts to provide synthetic and useful data. Either Tables or Figures are clear. The authors have provided both quantitative and qualitative results and have avoided redundancies. 

Discussion: “Discussion” are clear and comprehensive. The authors have developed critical outcomes based on previous results. 

I suggest the authors adding some more future research directions. 

Our Response: Thank you. The manuscript now discusses future research and policy directions. The conclusion now ends with the following:

“Future research directions include evaluating the influence of seasonality on food availability and access in the natural food environment during food system shocks and surveying additional geographical areas and informal settlements in Kenya. Future policies and interventions should focus on increasing household access to the natural food environment when available to mitigate the impact of food system shocks, particularly for informal settlements. This can be accomplished by partnering urban agricultural initiatives with abandoned lots and grounds of local establishments and businesses such as schools and churches. Furthermore, increasing food availability and access during food system shocks can mitigate food insecurity during these times. Lines 632-641

Reviewer 3 Report

The paper submitted for review deals with the extremely important issue of the impact of the COVID-19 pandemic on food security in Kenya.

Overall, the article is very interesting and presents an important cognitive layer. Nevertheless, with the addition of a few important aspects, it could be slightly better.

First, the purpose of the paper should be clearly stated at the end of the introduction. I also suggest at this point to include research questions or hypotheses that will facilitate reading s of the text.

In the methodological section, I missed information on how the authors took care of the quality of the telephone interviews with respondents, whether the interviews were conducted under homogeneous conditions (?), whether they were conducted by one person (?), whether the dynamics of the interview were adjusted to the interviewee (?), whether methods were used to exclude possibly falsified data (?). Telephone interviews are often subject to higher error, but I see that the authors have rightly noted this in the weaknesses and strengths of the study.

I suggest that throughout the text and tables, the notation of the p-value be corrected to lowercase "p", not "P".

Why don't the Authors brag about the strengths of their study?

Kind regards!

Author Response

The paper submitted for review deals with the extremely important issue of the impact of the COVID-19 pandemic on food security in Kenya.

Overall, the article is very interesting and presents an important cognitive layer. Nevertheless, with the addition of a few important aspects, it could be slightly better.

First, the purpose of the paper should be clearly stated at the end of the introduction. I also suggest at this point to include research questions or hypotheses that will facilitate reading s of the text.

Our Response:  Thank you. The study purpose has been clearly defined to ensure greater clarity. Line 85-86 now reads: “The purpose of this study is to inform policy on appropriate response measures to withstand future and potentially perpetual food system shocks.”

And, we also added the study’s primary hypothesis into the introduction, as shown below:

“We hypothesize that during the COVID-19 pandemic, urban residents will report decreased access to both the natural and built food environment negatively impacting food security.”  Lines 97-101

In the methodological section, I missed information on how the authors took care of the quality of the telephone interviews with respondents, whether the interviews were conducted under homogeneous conditions (?), whether they were conducted by one person (?), whether the dynamics of the interview were adjusted to the interviewee (?), whether methods were used to exclude possibly falsified data (?). Telephone interviews are often subject to higher error, but I see that the authors have rightly noted this in the weaknesses and strengths of the study.

Our Response:  Thank you. We took great care to highlight specific limitations. In response to your suggestion, we expanded upon data collection methods in the section “Data collection” which has now been modified as follows:

“The survey tool was incorporated into an online survey platform for ease of collection and analysis. The survey instrument was translated from English into Swahili by a member of the research team (N.M.). In both study settings, a study member (N.M.) called each selected HH to schedule a date and time to conduct the telephonic survey. Data was collected by four individuals, one member of the study team (N.M.) and three trained enumerators.  The enumerators conducted the survey in a standard setting (either an office at Moi University or AMPATH); however, it was not required that interviewees be in a standard setting. To keep methods consistent between enumerators, the interview was not adjusted for the interviewee outside the use of probing questions as dictated by the survey tool. To correct for potentially falsified data, the data were routinely audited, and outliers were removed from analysis.” Lines 185-195

I suggest that throughout the text and tables, the notation of the p-value be corrected to lowercase "p", not "P".

Our Response:  We agree. The p-value notation was corrected as suggested.

Why don't the Authors brag about the strengths of their study?

Our Response: Thank you but that is always a bit awkward as we are proud of the study and its findings particularly as the study was done under a most challenging time and environment. We anticipated the data and results would speak for themselves. The following was added to the Limitations section:

“While this study has several strengths such as study design and ascertaining new information relative to the influence the COVID-19 pandemic on food security in urban and rural communities in Kenya, there were also limitations.” Lines 551-553